# *Aspergillus flavus* and Total Aflatoxins Occurrence in Dairy Feed and Aflatoxin M_1_ in Bovine Milk in Aguascalientes, Mexico

**DOI:** 10.3390/toxins14050292

**Published:** 2022-04-20

**Authors:** Fernanda Álvarez-Días, Barenca Torres-Parga, Arturo Gerardo Valdivia-Flores, Teódulo Quezada-Tristán, José Isidro Alejos-De La Fuente, Joaquín Sosa-Ramírez, Erika Janet Rangel-Muñoz

**Affiliations:** 1Centro de Ciencias Agropecuarias, Universidad Autónoma de Aguascalientes, Aguascalientes 20131, Mexico; al266235@edu.uaa.mx (F.Á.-D.); barenca.torres@gmail.com (B.T.-P.); teodulo.quezada@edu.uaa.mx (T.Q.-T.); jsosar@correo.uaa.mx (J.S.-R.); janet.rangel@edu.uaa.mx (E.J.R.-M.); 2Departamento de Zootecnia, Universidad Autónoma Chapingo, Texcoco 56230, Mexico; jalejosd@chapingo.mx

**Keywords:** *Aspergillus flavus*, aflatoxins, aflatoxin M_1_, dairy cows, Mexican Highland Plateau, totally mixed rations

## Abstract

Contamination of food chains by toxigenic fungi and aflatoxins is a global problem that causes damage to human health, as well as to crop and livestock production. The objective is to evaluate *Aspergillus flavus* and total aflatoxins (AFs) occurrence in totally mixed rations (TMRs) for dairy cows and aflatoxin M_1_ (AFM_1_) in milk for human consumption. Ninety-nine dairy production units located in Aguascalientes, Mexico, were randomly selected, and samples were collected from TMRs, raw milk, and milk marketed in the city in two consecutive agricultural cycles. AFs were quantified in TMRs and milk by indirect enzyme immunoassay and HPLC; aflatoxigenic and molecular (PCR) capacity of monosporic *A. flavus* isolates in the feed was characterized. All feed, raw, and pasteurized milk samples showed aflatoxin contamination (26.0 ± 0.4 µg/kg, 32.0 ± 1.0, and 31.3 ± 0.7 ng/L, respectively), and a significant proportion (90.4, 11.3, and 10.3%) exceeded the locally applied maximum permissible limits for feed and milk (20.0 µg/kg and 50 ng/L). Aflatoxin contamination in both TMRs and milk indicated a seasonal influence, with a higher concentration in the autumn–winter cycle when conditions of higher humidity prevail. The results obtained suggest the existence of contamination by aflatoxigenic *A. flavus* and aflatoxins in the diet formulated for feeding dairy cows and, consequently, in the dairy food chain of this region of the Mexican Highland Plateau.

## 1. Introduction

*Aspergillus* spp. is a genus of telluric filamentous fungi very abundant on the planet. Most agricultural products for human or livestock intake are susceptible to contamination by these fungi during the various stages of cultivation, harvesting, storage, and transport [1]. Especially *A. flavus* has been associated with aflatoxins contamination in many different feed ingredients for dairy cows feeding [2,3].

Aflatoxins (AFs) are synthesized when *A. flavus* grows on grains, forages, and other substrates [4]. AFs are considered the most dangerous mycotoxins because of their great potential for liver, immune, and kidney damage [5,6]; moreover, chronic exposure to AFs induces mutagenic, carcinogenic, and teratogenic effects [7,8]. Exposure to AFs represents a public health problem, as it is claimed that 20–50% of all cancers are related to dietary factors [9,10]. AFs also affect animal health and lead to a decrease in milk production, decreasing the profitability of dairy production units [1,11,12].

It has been reported that the global problem of AF contamination is more severe in tropical and subtropical climates [13]. In addition, evidence of AF contamination in dairy cow rations has been found in studies carried out in several countries [14,15,16]. In Mexico, it has been estimated that the natural frequency of AFs occurs widely (99.3%) in dairy cow feed, which has been associated with the presence (39.9%) of milk contamination by a hydroxylated derivative of AFs called aflatoxin M_1_ (AFM_1_) [17]. Thus, the presence of AFs in dairy cow feed represents a risk to human health by a carry-over effect since a fraction (1.0–6.0%) of ingested AFs are excreted in milk as AFM_1_ [18].

Although multiple studies have been conducted on the food contamination scenario by *A. flavus* and AFs, the information available in the Mexican Highland Plateau region is scarce and dispersed. Therefore, the objective of this study was to evaluate the contamination by toxigenic strains of *A. flavus* and AFs of the total mixed ration (TMR) for dairy cows, as well as the risk of contamination by AFM_1_ in the milk locally marketed in the Mexican Highland Plateau.

## 2. Results and Discussion

### 2.1. Fungi Isolation

Slightly less than half (48.4%) of the feed samples obtained from the dairy production units (DPUs) were found to contain a fungal concentration greater than 10^6^ CFU/kg, while 10.1% showed a count greater than 10^8^ CFU/kg. In the feed samples, the growth of one or more fungal colonies was observed in 88.9% of the 99 participating DPUs. A total of 2496 fungal isolates were identified, of which 55.5% showed morphological characteristics coinciding with those described for the *Aspergillus* genus. Isolates with morphology corresponding to that of the genera *Mucor, Fusarium, Cladosporium, Rhizopus, Penicillium, Emericella,* and *Candida* were also found (21.0, 14.1, 3.6, 2.9, 1.6, 0.9, and 0.3%). Of the *Aspergillus spp*. isolates, 1.5% (21/1385) corresponded to *A. flavus* morphology. These proportions are consistent with those found in studies carried out in Mexico, where the same fungal genera were identified; likewise, moderate to high fungal contamination counts (higher than 10^6^ CFU/kg) have been found in feed and soil [17,19,20]. Only 19 *A. flavus* strains obtained from dairy cow feed produced AF (10.4 ± 1.7 mg/kg) in the culture media.

### 2.2. Molecular Characterization

Twenty-one TMR isolates congruent with the morphology of *A. flavus* and with the capacity to produce a detectable concentration of AFs in the culture media were obtained. All these isolates amplified the internal transcribed spacers region (ITS), the calmodulin gene (CaM), and the aflatoxin biosynthetic pathway gene (aflR). Therefore, they were identified as *A. flavus*, considering that the International Fungal Barcoding Subcommittee has proposed the ITS region as the default barcode for the identification of these fungi. Additionally, it has been pointed out that the CaM gene provides better resolution to identify most *Aspergillus* species [21], while the expression of the aflR gene is related to the AF production capacity of *A. flavus* isolates [22].

### 2.3. Quantification of Aflatoxins in Total Mixed Ration

AF contamination was detected in all TMR samples obtained from participating DPUs. The mean AF concentration was 26.0 ± 0.4 µg/kg. Of all samples, 90.4% exceeded the maximum permissible limit (MPL) of AFs for feed intended for dairy cows recommended by legislation in Latin America and the U.S. (20 µg/kg) [14], although all samples exceeded the current MPL for AF content in rations in the European Union (5 µg/kg). Other studies carried out in Mexico have also detected comparable values of AF contamination in both concentrated feeds and dairy cattle feed [17,19,23]. The formulation of the ration fed for the dairy cows in each of the DPUs of the study was very heterogeneous since different ingredients and different proportions were used, and the origin of the foodstuffs was from states of the Mexican Highland Plateau. When concentrates or protein nuclei were included in the total mixed rations, there was a higher AF concentration (*p* < 0.05) compared with when they were not included (Table 1). The range of AF contamination in the samples with concentrates was not wide (25.9–27.0 µg/kg), suggesting that the quality and composition of the concentrates were homogeneous; however, the proportion of samples exceeding the MRL was very large (87.5–100%). This may be because the main ingredients (cereals, oilseeds, flours, by-products) from which concentrate feeds or commercial protein nuclei are made are very susceptible to contamination by toxigenic fungi and, therefore, mycotoxins and come from different geographical origins [24].

The average concentration of AFs in the TMR samples obtained in the autumn–winter period was significantly higher (*p* < 0.05) compared with the estimated AF concentration in the spring–summer period (Table 2). Similarly, a significant association (χ^2^) was detected between samples collected in autumn–winter and the proportion of samples that exceeded the MRL (*p* < 0.05), so the risk (odds ratio) of finding feed with concentrations above the MRL was more than double that of the spring–summer period.

Coincidentally, the average AFM_1_ content in milk obtained in the autumn–winter period was significantly higher (*p* < 0.05) compared with AFM_1_ in samples collected in the spring–summer period (Table 2). Moreover, there was a significant association (χ^2^: *p* < 0.01) in the proportion of milk samples that exceeded the MPL (50 ng/L) in autumn–winter [25], so the odds ratio of finding milk with concentrations above the MPL was more than three times higher than in the –summer period.

These data suggest that there was a seasonality effect in the level of AFM_1_ in milk and in the higher proportion of samples that exceeded the MPL; therefore, the season of the year with the highest contamination in milk coincided with the season of the highest presence of AFs in the rations. AFM_1_ contamination has been reported in several countries around the world at levels that exceed the maximum permissible limit of European legislation [25]. Similar studies in Jordan, Iran, Italy, Brazil, Nicaragua, and Mexico have also found a seasonality effect, with winter being the season with the highest AFM_1_ concentration in dairy products [26,27,28,29,30,31]. In this study, it was found that a high percentage (92%) of the samples with fresh forages (Table 1) had a high AF content, which coincides with the suggested idea that the result could be attributed to both the winter shortage of feedstuffs and a suggestion that the weather conditions prevailing in that season cause stress on toxigenic fungi, especially contaminating feedstuffs stored under inadequate conditions [32,33].

The results of this study show an association of AF contamination as a function of the climatic conditions prevailing during the crop growing and storage cycle of agricultural products, which has been attributed to the dependence of *A. flavus* on optimal conditions of water activity and ambient temperature (0.96–0.98 aw, 28–30 °C) for AF production [34,35]. The effect of time of year on aflatoxin contamination in dairy cattle feed in several countries has been described previously [17,27,33]. In the Mexican Highland Plateau, the highest agricultural production occurs during the spring–summer cycle, so both grains and fodder produced in that season go into storage and are distributed and used in the autumn–winter season [36]. AF contamination of stored agricultural products may be due to *A. flavus* causing infection in the field, as well as reinfection in storage when it finds optimal conditions of temperature and humidity [33]. This study found that most feed samples (90.4%) had AF contamination above the MPL applied in all the countries (20 µg/kg), regardless of their geographic origin (Table 3), suggesting that elevated contamination of raw materials was associated with TMR contamination as a widespread geographic event.

### 2.4. Quantification of Aflatoxins in Pasteurized Milk

The presence of AFM_1_ was found in all of the pasteurized milk samples marketed in city areas of the state of Aguascalientes; the samples registered AFM_1_ levels in a range of 10.6–73.8 (ng/L), while the mean concentration was 30.9 ± 6.0 (ng/L), and 10.3% of the samples surpassed the MPL established by the European Union (50 ng/L) [25]. Although this concentration represents a risk to human health, the milk can be marketed because the MRL allowed by legislation in Mexico and all nations that follow the Codex Alimentarius tolerates a tenfold higher content of AFM_1_ in milk than the European standard [13]. In this study, a significant difference (*p* < 0.05) was observed between the means of AFM_1_ concentration in milk marketed in the city as a function of the place of origin of the milk (Table 4); however, it was not related (*p* > 0.5) to the origin of the feed (Table 3). In a similar study that evaluated AFM_1_ contamination in pasteurized and ultrapasteurized milk of different brands, the authors found aflatoxin contamination in milk from states located in the Mexican Highland Plateau [37].

## 3. Conclusions

The results of the study indicated that contamination of the dairy food chain of the Mexican Highland Plateau is frequent through the occurrence of aflatoxigenic *Aspergillus flavus* and its aflatoxins, both in agricultural products and in the persistence of their hydroxylated metabolites in raw or pasteurized milk destined for human consumption. This fact is influenced by both the conditions prevailing in each agricultural cycle and the exchange of feed and traded milk among the different states that make up this biogeographic region. The results suggest that it is necessary to design more effective strategies to verify the safety of the ingredients used in feed formulation, with the purpose of offering safe dairy products to the consumer, as well as maintaining animal health and achieving the potential productivity of dairy cows.

## 4. Materials and Methods

### 4.1. Study Design

The study of the contamination of the dairy food chain in the Mexican highland plateau (MHP) was developed in the following three stages: (a) AF and *A. flavus* in TMR, (b) AFM_1_ in raw milk, and (c) AFM_1_ in milk marketed in urban areas. We selected dairy production units (DPUs) registered in the official registers (N = 3155) located in the biogeographic province of the MHP, which granted permission to obtain feed and raw milk samples and provided the reference data. A sample size (*n* = 93 + 6 possible defections) to estimate proportions was calculated for a finite population (without replacement), considering a precision of 10%, 95% confidence level, and an expected proportion of 0.5 of DPUs with TMR or milk contaminated by AF or AFM_1_. Samples of TMR and milk were collected in 99 DPUs in two consecutive agricultural cycles (spring–summer and fall–winter 2020–2021). Additionally, samples of commercially sold milk (pasteurized and ultrapasteurized) were acquired by visiting each establishment included in an alphabetically ordered list of supermarkets in the state of Aguascalientes. The method of collecting commercial milk samples was nonprobabilistic in a “snowball” style, in which the acquisition of new milk samples was suspended when in three consecutive establishments no brands or origins different from those previously acquired were located; their origin was also recorded (*n* = 8 origins, 27 brands, 170 samples).

### 4.2. Sample Collection

TMR samples without mycotoxin binders, antioxidants, or fungal inhibitors were collected directly from the mixer wagon, identifying five “M” points from which a sample (1.0 kg) was taken for each site. The composite samples (5.0 kg) were placed in a plastic bag, homogenized, and a subsample (1.0 kg) was obtained. The samples were transported to the laboratory (1–4 °C); they were dried in a forced-air oven, ground, and kept frozen (−20 °C) until analysis (<2 weeks). Samples (0.6 L) of raw milk were obtained directly from the collecting tank and corresponded to two milkings per day, while pasteurized milk (1.0 L) was obtained from the display rack; refrigerated milk samples were carried to the laboratory and kept frozen (−20 °C; <1 week). The milk samples were skimmed by centrifugation (10 min/3500 s/10 °C), and the defatted supernatant was tested. During sampling, a data collection instrument was used to collect data from the managers of the dairy farms and retailers, and a database was created with zootechnical and commercial information on the origin of the TMR and milk samples.

### 4.3. Characterization of Fungal Isolates

The total count of fungal colonies in the feed was performed using the serial dilution plating technique [19]. The TMR samples were diluted (10^−1^, 10^−2^, 10^−3^, and 10^−4^), and sowings were performed on rose bengal agar + chloramphenicol and Czapeck, incubated in the dark (27–30 °C for seven days). Wet preparations of fungal colonies were made with cotton blue-lactophenol [38]. Initial identification of *A. flavus* isolates was based on their macroscopic and microscopic characteristics [39,40]. AF concentration in monosporic cultures (culture media plus mycelial mass) was estimated by HPLC on a dry matter basis, as explained below for feed samples.

### 4.4. Identification of Aspergillus flavus Isolates by PCR

Genomic DNA was obtained from monosporic isolates by standardized methods [41,42,43]. DNA quality was visualized by agarose gel electrophoresis (1.0%). DNA samples were deposited on the gel with loading buffer (PlatinumTM II Green PCR Buffer 5X, Thermo Fisher Scientific, Waltham, MA, USA) and placed in a buffered electrophoretic chamber (TAE 1x, 95 volts, 40 min); subsequently, banding was visualized under UV light in a photodocumenter (Bio-Rad Molecular Imaging^®^-Gel Doctm XR, Hercules, CA, USA) with Quantity One software (version 4.6.7). Using polymerase chain reaction (PCR), a fragment of the internal transcript spacers (ITS1-5.8S-ITS2-ITS2-rRNA), the calmodulin gene (CaM), and the aflatoxin biosynthetic pathway initiator gene (aflR) were amplified based on previously described protocols [44,45,46]. The following primers were used for ITS1, ITS4, CaM-F, CaM-R, aflR-F, and aflR-R: 5′-TCCGTAGGTGAACCTCTGCGG-3′, 5′-TCCTCCGCTTATTGATATATG-3′, 5′-GCCAAAATCTTCATCCGTAG-3′, 5′-ATTTCGTTCAGAATGCCAGGCAGG-3′, 5′-GGATAGCTGTACGAGTTGTGCCAG-′3 and 5′-TGGKGCCGCCGACTCGAGGAAYGGGT-3′ (Thermo Fisher Scientific, Waltham, MA, USA). A ladder of molecular weight marker (1.0 μL, DNA Ladder 100 bp, 0.5 µg/ μL, Thermo Fisher Scientific) and buffer (1.0 μL, BlueJuice gel loading buffer 10X, Thermo Fisher Scientific) were included in the first lane of the gel. PCR products were purified with ExoSAP-IT^®^ PCR Product Cleanup reagent (Afflymetrix, Thermo Fisher Scientific Inc., Santa Clara, CA, USA).

### 4.5. Quantification of Aflatoxin Concentration

Quantification of total aflatoxins (B_1_, B_2_, G_1_, and G_2_) from TMR was performed according to AOAC official method 990.33 [47]; AF content in TMR samples was extracted in solid-phase tubes (Supelclean LC-CN, Supelco, CA, USA). The extracts were processed with trifluoroacetic acid and injected into an HPLC system (Varian Associates Inc., Sydney, Australia) with a fluorescence detector. Estimation of AF concentration was performed with the aid of software (Ridasoft Win ver. 1.8) and by comparing against a calibration curve prepared with purified aflatoxins (B_1_, B_2_, G_1_, and G_2_; Sigma Aldrich, St. Louis, MO, USA). The limit of quantification of the HPLC method for AF was 2.5 µg/kg.

AFM_1_ quantification in milk samples was performed by indirect enzyme immunoassay [5,14,16]. The milk samples were skimmed by centrifugation (10 min/3500 s/10 °C), and the defatted supernatant was tested. The extraction of AFM_1_ was performed with methanol (70%) according to the manufacturer’s instructions (Ridascreen Fast^®^ Aflatoxin and Aflatoxin M1, R-Biopharm, Darmstadt, Germany, ref. R1121). The absorbance reading was performed in a microplate reader (ELx800TM, Bio Tek, Winooski, VT, USA) at 450 nm, and the results were compared to a calibration curve prepared with purified AFM_1_ (Sigma Aldrich, St. Louis, MO, USA).

### 4.6. Statistical Analysis

Quantitative data were subjected to analysis of variance (ANOVA) using the general linear model (GLM) procedure in the Statistical Analysis System (2004) software. To identify significant differences in the categorical variables (samples exceeding or not exceeding the LMP), the Chi-square test (χ^2^) was performed. A probability level of *p* < 0.05 was considered. The odds ratio was calculated using a ratio of the DPU portion that exceeded the MPL for AF concentration and were exposed to a specific factor (a particular feedstuff, sampling time, or milk origin) divided by the DPU portion that exceeded the MPL for AF concentration but was not exposed to that specific factor.

## Figures and Tables

**Table 1 toxins-14-00292-t001:** Odds ratio (OR) associated with the frequency of samples exceeding the maximum permissible limit (MPL) of aflatoxin (AF) concentration, according to the inclusion of feedstuffs in totally mixed rations used in dairy production units (DPU).

Feedstuff	DPU (*n*)	Incorporated into TMR			Not Incorporated into TMR			P(χ^2^)	OR
		AF (µg/kg ± SEM)	CI 95%	>MPL (%)	AF (µg/kg ± SEM)	CI 95%	>MPL (%)		
Fresh forage	81	26.3 ^a^ ± 0.4	25.7–26.9	92.0	24.8 ^a^ ± 0.8	23.6–26.0	83.3	0.11	2.29
Grain	41	26.3 ^a^ ± 0.6	25.5–27.1	92.7	25.8 ^a^ ± 0.5	25.2–26.5	88.8	0.36	1.6
Agro-industrial by-product	21	25.2 ^a^ ± 0.8	24.1–26.3	90.5	26.3 ^a^ ± 0.4	25.7–26.8	90.4	0.99	1.0
Corn silage	88	25.9 ^a^ ± 0.4	25.4–26.5	90.3	26.9 ^a^ ± 1.1	25.4–28.4	90.9	0.93	0.9
Concentrate	78	26.5 ^a^ ± 0.4	25.9–27.0	89.1	24.4 ^b^ ± 0.8	23.3–25.5	95.2	0.23	0.4
Straw	89	25.9 ^a^ ± 0.4	25.4–26.4	89.3	27.2 ^a^ ± 1.1	25.6–28.8	100	0.12	0.0
Total	99	26.0 ± 0.4	25.3–26.7	90.4					

MPL—maximum permissible limit 20 µg/kg; ^ab^—means in the same row with different literals show statistically significant differences (*p* < 0.05; Tukey’s HSD test). Abbreviations: SEM—standard error of the mean; CI 95%—Confidence Interval at 95%; P(χ^2^)—probability of the chi-square test (*p* < 0.05).

**Table 2 toxins-14-00292-t002:** Odds ratio (OR) associated with the frequency of feed and milk samples exceeding the maximum permissible limit (MPL) of aflatoxin (AF) or AFM_1_ concentrations, according to sampling time in dairy production units (DPU).

Sampling Time	Mean ± SEM	CI 95%	>MPL (%)	P(χ^2^)	OR
**Totally mixed rations (AF µg/kg)**					
Fall-winter	26.9 ^a^ ± 0.4	26.3–27.5	93.4	0.03	2.87
Spring-summer	24.1 ^b^ ± 0.6	23.2–25.0	83.3	0.03	
Total	26.0 ± 0.4	25.3–26.7	90.4		
**Raw milk (AFM_1_ ng/L)**					
Fall-winter	36.1 ^a^ ± 0.8	34.0–38.4	16.9	0.00	3.5
Spring-summer	27.8 ^b^ ± 1.2	26.3–29.4	5.6	0.00	
Total	32.0 ± 1.0	30.2–33.9	11.3		

MPL—maximum permissible limit in feed (20 µg/kg) and raw milk (50 ng/L). ^ab^—means in the same column with different literals show statistically significant differences (*p* < 0.05; Tukey’s HSD test). Abbreviations: SEM—standard error of the mean; CI 95%: LL and UL—lower and upper confidence intervals at 95%; P(χ^2^)—probability of the chi-square test (*p* < 0.05).

**Table 3 toxins-14-00292-t003:** Odds ratio (OR) associated with the frequency of dairy rations exceeding the maximum permissible limit (MPL) of aflatoxin (AF) concentration by the origin of feedstuffs included in totally mixed ration in dairy production units (DPUs).

Origin of Feedstuffs (State)	DPU (*n*)	AF (µg/kg)	±SEM	CI 95%	>MPL (%)	P(χ^2^)	OR
Durango	12	27.0 ^b^	±1.0	25.5–28.4	87.5	0.61	1.4
Zacatecas	36	25.9 ^b^	±0.6	25.1–26.8	93.1	0.34	0.6
Jalisco	51	26.3 ^b^	±0.5	25.6–27.0	95.1	0.02	0.3
San Luis Potosí	2	32.1 ^a^	±2.5	28.6–35.6	100	0.51	0.0
Michoacán	4	26.6 ^b^	±1.8	24.1–29.2	100	0.35	0.0
Guanajuato	5	25.6 ^b^	±1.6	23.3–27.8	100	0.29	0.0
Aguascalientes	99	26.0 ^b^	±0.4	25.3–26.7	90.4	-	-
Total	99	26.0	±0.4	25.3–26.7	90.4		

MPL—maximum permissible limit in feed (20 µg/kg) and raw milk (50 ng/L). ^ab^—means in the same column with different literals show statistically significant differences (*p* < 0.05; Tukey’s HSD test). Abbreviations: SEM—standard error of the mean; CI 95%: LL and UL—lower and upper confidence intervals at 95%; P(χ^2^)—probability of the chi-square test (*p* < 0.05).

**Table 4 toxins-14-00292-t004:** Odds ratio (OR) associated with the frequency of marketed milk surpassed the maximum permissible limit (MPL) of aflatoxin M_1_ concentration (AFM_1_), according to the origin of the pasteurizer brand.

Origin of Milk	Samples (*n*)	AFM_1_ (ng/L)	CI 95%	SEM	>MPL (%)	P(χ^2^)	OR
State of Mexico	20	37.3 ^a^	34.6–40.1	±2.4	20.0	0.03	2.52
Aguascalientes	33	34.9 ^ab^	32.8–37.0	±1.7	16.7	0.06	2.08
Hidalgo	10	32.1 ^abcd^	28.2–36.0	±2.9	10.0	0.96	0.97
Durango	41	31.9 ^bc^	29.9–33.8	±1.5	12.2	0.52	1.29
Jalisco	30	28.9 ^cd^	26.7–31.2	±1.2	3.3	0.05	0.26
Nuevo León	15	27.0 ^cde^	23.8–30.1	±2.3	6.7	0.49	0.60
Guanajuato	18	25.6 ^de^	22.7–28.5	±1.5	0.0	0.03	0.0
San Luis Potosí	3	18.1 ^e^	11.0–25.2	±0.7	0.0	0.40	0.0
Mean	170	31.3	29.9–32.6	±0.7	10.3		

MPL—maximum permissible limit in milk (50 ng/L). ^abcde^—means in the same column with different literals show statistically significant differences (*p* < 0.05; Tukey’s HSD test). Abbreviations: SEM—standard error of the mean; CI 95%/LL and UL—lower and upper confidence intervals at 95%; P(χ^2^)—probability of the chi-square test (*p* < 0.05).

## Data Availability

Not applicable.

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
