# Peer review of "Aspergillus flavus and Total Aflatoxins Occurrence in Dairy Feed and Aflatoxin M1 in Bovine Milk in Aguascalientes, Mexico"

_toxins, 2022, doi:10.3390/toxins14050292_

Round 1

Reviewer 1 Report

Manuscript ID: toxins-1660543: Contamination of dairy food chain by Aspergillus flavus and aflatoxins in Aguascalientes, Mexico.

This work describes the contamination animal rations with fungi and aflatoxins, including the feedstuffs of different origins; and the occurrence of aflatoxin M1 in milk is studied as well.

This work is well conducted and the manuscript is well written.

However, similar works were previously reported in literature, particularly in Mexico as indicated by the authors in L66-70.

The following comments and suggestions are addressed to the authors:

L116: the contents of AF in milk obtained during autumn-winter are higher than that of spring-summer period. In addition, 92% of fresh forage exceeded the MPL of contamination with AF (Table 1). So that, the contamination with AF is not only due to storage conditions, which can explain the contents of AFM1 in milk higher than that of TMR (Table 2). What do you think of this idea.

Table 3: (origin of feedstuffs and their AF contents) is not commented in the body of the text: (The feedstuff samples are mostly (87.5%-100%) contaminated with high contents of AF independently of their origin, which lead to the contamination of TMR). What do you think.

L156: “AFM1 was detected in 100% of the pasteurized milk samples marketed in urban areas”. So that, this milk should not be consumed. What do you think.

L2: aflatoxin or aflatoxins: do you mean total aflatoxins. Please verify that.

L7: aflatoxin M1 instead of AFM1.

L15, L72, L82, L217: TMR instead of RTM

L131 (title of Table 2): eliminate “according to the”

L151, L169: ng/L instead of ng/kg

Author Response

This work is well conducted and the manuscript is well written. However, similar works were previously reported in literature, particularly in Mexico as indicated by the authors in L66-70.

L66-70

The text was modified to explain that the proportions of fungal genera detected in our work are comparable with those found in other studies carried out in Mexico

The contents of AF in milk obtained during autumn-winter are higher than that of spring-summer period. In addition, 92% of fresh forage exceeded the MPL of contamination with AF (Table 1). So that, the contamination with AF is not only due to storage conditions, which can explain the contents of AFM1 in milk higher than that of TMR (Table 2). What do you think of this idea.

L116:

The text was modified to explain that the result could be attributed to both the winter shortage of feedstuffs and that the weather conditions prevailing in that season cause stress on toxigenic fungi, especially contaminating feedstuffs stored under inadequate conditions.

Is not commented in the body of the text: (The feedstuff samples are mostly (87.5%-100%) contaminated with high contents of AF independently of their origin, which lead to the contamination of TMR). What do you think.

Table 3

It was explained in the text that the proportion of samples exceeding the MRL was very large (87.5-100%). This fact may be because the main ingredients from which the concentrated feeds are made are very susceptible to mycotoxin contamination, which seems to be independent of the geographical origins, within the Mexican Highland Plateau.

 “AFM1 was detected in 100% of the pasteurized milk samples marketed in urban areas”. So that, this milk should not be consumed. What do you think.

L156:

Although this concentration represents a risk to human health, the milk can be marketed because the MRL allowed by legislation in Mexico (500 ng/L) and in all nations that follow the Codex Alimentarius, tolerates a tenfold higher content of AFM1 in milk than the European standard (50 ng/L)

Aflatoxin or aflatoxins: do you mean total aflatoxins. Please verify that.

Line 2:

Title was rewritten

Aflatoxin M1 instead of AFM1.

Line 7:

Aflatoxin M1 was written instead of AFM1.

TMR instead of RTM

Lines 15, l72, l82, l217:

The abbreviation TMR was rewritten in place of RTM throughout the document.

Eliminate “according to the”

Line 131 (title of Table 2):

The expression "according to" duplicate in title of Table 2 was removed

ng/L instead of ng/kg

Lines 151, l169:

Was written ng/L instead of ng/kg throughout the document

Reviewer 2 Report

This article has serious flaws, and additional experiments should be carried out or correctly described:

The materials and methods section needs to be improved by including information on the extractions of all types of samples analysed and including the correct validation process for mycotoxin detection. To sum up, this section is confusing and lacks crucial information for the methodology that converts the article into unreproducible and is challenging to revise appropriately. Therefore, in the present form, the article does not adequately describe the applied methodology and cannot support the results contained therein.

There are other flaws throughout the work but I think the authors should start the corrections with the materials and methods.

Author Response

The materials and methods section was improved by including information on the extractions of all types of samples analyzed and including the approved method for the detection of mycotoxins.

The similarity index was decreased (35 to 10%).

Reviewer 3 Report

The authors sampled feed, raw milk and marketed milk in two seasons and check for presence of aflatoxigenic A. flavus and aflatoxin contamination. The manuscript is well-written and describes good, clean work that yields important information about the frequency of mycotoxin contamination in a food production chain. Readers will appreciate that the authors compare their results to similar studies and to local maximum allowable limits.

I recommend that the authors soften the statement at lines 14-16 by using a verb like ‘indicated’ or ‘suggested’ instead of ‘showed’. While similar results are observed elsewhere, the current work only tested each season once (not multi-year).

Minor comments/suggestions:

line 33- italicize A. flavus

line 83- make red text black

all tables- format columns to have same format (ex. CI 95% as one column instead of two for LL and UL)

line 204-205- sample mass should be larger than subsample mass

line 268- there is an extra comma

RTM is used in 4 places, was TMR meant? Otherwise, it should be defined.

Author Response

I recommend that the authors soften the statement at lines 14-16 by using a verb like ‘indicated’ or ‘suggested’ instead of ‘showed’. While similar results are observed elsewhere, the current work only tested each season once (not multi-year).

Lines 14-16

The verb 'showed' was replaced by 'indicated'

Italicize A. flavus

Line 33

A. flavus was italicized  

Make red text black

Line 83

The red text was made color black

Format columns to have same format (ex. CI 95% as one column instead of two for LL and UL)

All tables

Only one format column was used

Sample mass should be larger than subsample mass

Lines 204-205

Information on the composite sample and the subsample was supplemented

There is an extra comma

Line 268

Extra comma was removed

RTM is used in 4 places, was TMR meant? Otherwise, it should be defined.

RTM was replaced by TMR at entire document.

Round 2

Reviewer 2 Report

The authors have revised the article and improved its quality.

Minor corrections:

  • > 0.05 instead of P > 0.05 or 0.5.
  • A. Flavus not A. Flavus.
  • Please proofread the article to avoid these typing errors

Author Response

The content of this third version of the manuscript addresses each of the comments made in your review.

The authors will be attentive to any additional suggestions and comments that you may wish to make to our study.